# Perceived Value of Electronic Medical Records in Community Health Services: A National Cross-Sectional Survey of Primary Care Workers in Mainland China

**DOI:** 10.3390/ijerph17228510

**Published:** 2020-11-17

**Authors:** Zining Xia, WenJuan Gao, Xuejuan Wei, Yingchun Peng, Hongjun Ran, Hao Wu, Chaojie Liu

**Affiliations:** 1Fangzhuang Community Health Service Center of Capital Medical University, Beijing 100078, China; xznlxy826@gmail.com (Z.X.); b6697238@gmail.com (W.G.); weibing0918@gmail.com (X.W.); pycjql@ccmu.edu.cn (Y.P.); rhj1030@gmail.com (H.R.); 2School of Psychology and Public Health, La Trobe University, Melbourne 3086, Australia

**Keywords:** electronic medical record, primary care, China

## Abstract

Objective: To evaluate the degree to which electronic medical records (EMRs) were used in primary care and the value of EMRs as perceived by primary care workers in China. Methods: A cross-sectional survey was conducted on 2719 physicians (*n* = 2213) and nurses (*n* = 506) selected from 462 community health centres across all regions of mainland China except for Tibet. Regional differences in the responses regarding the functionality of existing EMR systems and the perceived value of EMRs were examined using Chi-square tests and ordinal regression analyses. Results: Less than 59% of the community health centres had adopted EMRs. More than 89% of the respondents believed that it was necessary to adopt EMRs in primary care. Of the existing EMR systems, 50% had access to telehealth support for laboratory, imaging or patient consultation services. Only 38.4% captured data that met all task needs and 35.4% supported referral arrangements. “Management of chronic conditions” was voted (66%) as the top preferred feature of EMRs. Higher levels of recognition of the value of EMRs were found in the relatively more developed eastern region compared with their counterparts in other regions. Conclusions: Rapid EMR adoption in primary care is evident in mainland China. The low level of functionality in data acquisition and referral arrangements runs counter to the requirements for “management of chronic conditions”, the most preferred feature of EMRs in primary care. Regional disparities in the realised value of EMRs in primary care deserve policy attention.

## 1. Introduction

Electronic medical records (EMRs) have been widely adopted and used in healthcare organisations. EMRs are computerised patient records introduced in the early 1970s as a way to better organise and use patient healthcare records [1]. They are usually kept within a healthcare delivery setting involving one or more encounters [2]. Many countries have provided financial incentives to encourage the adoption and use of EMRs [3]. EMRs can serve multiple purposes at the individual, organisational and system levels, such as clinical care for individual patients, billing, workforce planning, epidemiological and public health studies, health policy development and evaluations [4,5]. EMRs also serve as a legal document, which is critical in resolving patient complaints [6].

Extensive studies have been conducted to evaluate the benefits of EMRs. EMRs can help increase work efficiency, reduce medical errors, and improve communications and patient care outcomes [4,7]. A study in China found that 84% of medical workers believed that EMRs improved the quality of records, although 92% reported a shortened time spent in writing [8]. Empirical evidence shows that medical practitioners are more likely to adhere to evidence-based clinical guidelines under an EMR environment [9]. It was estimated that EMRs are associated with up to a 55% reduction in serious medication errors [10]. The overall operational performance of an organisation can also be enhanced as a result of the increased use of health information technologies [11]. The use of EMRs can result in a reduction of 13–27% of redundant laboratory tests. Societal benefits of EMRs include an improved ability to conduct research and the increased uptake of vaccinations against influenza and pneumococcal disease [12].

Despite strong evidence of the overall benefit, the adoption of EMRs can be hampered due to the high upfront acquisition and maintenance costs and temporary losses in productivity as a result of disruptions to workflows [9]. The primary care sector bears a high risk of lagging behind in the adoption of EMRs thanks to its large number and small scale. In early 2008, only a small minority (17%) of physicians reported using EMRs in their office setting in the US [13]. A study in seven countries concluded that the adoption of EMRs in primary care had failed to catch up with the increasing demands of patients resulting from obesity and population aging [14]. Although strong financial subsidies provided by various governments, such as in the UK [15] and Australia [16] can help achieve universal coverage of EMRs in primary care, most developing nations do not have the capacity to do so. In China, the central government developed a policy goal of establishing one community health centre for every 20,000 to 60,000 population as a strategy for strengthening primary care. However, the responsibility of financial support for these facilities has been delegated to local governments. This has led to significant disparities across regions in the development of community health services [17]. It is not clear how EMRs have been developed and used in community health services in China.

This study aimed to evaluate the degree to which EMRs were used in community health services and the value of EMRs as perceived by primary care workers in China. Most existing evaluative studies on EMRs have been conducted in the hospital sector [18]. There is a gap in the literature documenting the use of EMRs in primary care. The findings of studies on hospitals are hardly extrapolatable to the primary care sector. This is because the role of primary care in a health system is fundamentally different from that of hospitals. While hospital care is episodic, primary care is characterised by comprehensiveness and continuity, which also involves the coordination of care [19]. A recent systematic review identified the need to study the value of EMR use in primary care [1]. However, it is noted that very few EMRs in the world, if any, have achieved their full potential [20]. Unlike most existing studies that focus primarily on the use (or nonuse) of EMR components by health workers [3], this study aims to contribute to the literature by assessing the benefits EMRs can offer to patients in primary care. In China, the government has attempted to establish a hierarchical delivery system in which primary care providers serve as the first contact for patients, manage chronic conditions, refer patients for specialist and hospital care when needed, and coordinate patient care across services [21]. Indeed, primary care is most valued for its features in supporting the comprehensiveness, continuity and coordination of patient care [19]. The patient benefits assessed in this study derived from these features and included communication and work efficiency, coordination in clinical decisions, information and relational continuity, and quality of personalised care. EMR systems, regardless whether they are for primary care or hospital care, usually require some common components such as patient registration, documentation and charting, order entry, scheduling and reminders, assessment and alerts, and billing [22]. However, the daily use of these components does not tell what benefits patients obtain from the processes. The meaningful use of a range of EMR components in combination is often necessary to achieve the values that differentiate primary care from hospital care.

## 2. Methods

This study adopted a cross-sectional survey design. The study protocol was approved by the institutional review board of Fangzhuang Community Health Service Center of Capital Medical University.

### 2.1. Study Setting

A questionnaire survey of health workers was conducted in a nationwide sample of community health centres in mainland China, covering the four municipalities overseen directly by the central government and all (26) of the provinces except for Tibet. These provinces/municipalities are divided into three regions (http://www.stats.gov.cn/): 11 in the least developed western region (Sichuan, Chongqing, Guizhou, Yunnan, Shaanxi, Gansu, Qinghai, Ningxia, Xinjiang, Guangxi, Inner Mongolia), 11 in the most developed eastern region (Beijing, Tianjin, Hebei, Liaoning, Shanghai, Jiangsu, Zhejiang, Fujian, Shandong, Guangdong, Hainan), and 8 in the central region (Shanxi, Jilin, Heilongjiang, Anhui, Jiangxi, Henan, Hubei, Hunan).

The medical care delivery system in China is characterised by a hierarchical structure, comprising of 933,024 primary care institutions, 8422 secondary hospitals and 2340 tertiary hospitals (in 2017). Primary care institutions eligible for this study were those registered with the health commission under the code B100 (34,652 urban community health centres) and C100 (36,551 rural township health centres). Both were financed by the local governments under the community health services program according to their designated geographic catchment and the population size serviced. They also deliver individual-based medical care services covered by the social health insurance schemes. In this study, the urban community and rural township health centres were collectively referred to as the community health centres. The study sample did not include the large volume of small clinics, mostly privately-owned solo practices without EMRs. In 2017, the community health centres employed 664,252 (9.33 percent) registered doctors (including assistant doctors) and 503,084 (7.07 percent) nurses. They received 173 million outpatient visits (1.25 per capita), accounting for 21% of the total outpatient care services in China [23].

The integration of individual- and population-based health care services imposes a great challenge to the development of EMRs in community health centres. Unlike their hospital counterpart, in which services are restricted to the boundary of the organisation, community health services encompass the entire lifecycle of a person and have a population focus. The scope of services extends beyond the organisational boundary. In China, the central government has issued functionality standards for information systems in community health services, which covers outputs for various types of services, data requirements, and security and access control [24]. The standards acknowledge that it is the local governments that plan and provide financial support to the development of EMRs in community health centres. Therefore, the central government categorises the standards into essential, partly essential, and recommended requirements so that local governments can select the functional modules that are most appropriate to their socioeconomic status to tailor to the needs of the local residents.

### 2.2. Study Participants

Eligible participants of the survey were registered medical doctors (including assistant doctors) and nurses who had worked in the community health centres for no less than two years.

We estimated the sample size for this study using the formula *n* = u_a_^2^ σ (1 − σ)/δ^2^ [25]. A pilot study was conducted in Henan, one of the most populous provinces with socioeconomic status in the middle range of all provinces in China. The study found that in Henan province, 52% (12/23) of community health centres had established an EMR system. To allow a precision of δ = 0.05 and α being set at 0.05 (u_a_ = 1.96), we would need at least 384 participating centres to make an accurate estimation of the use of EMRs in community health centres: *n* = (1.96)^2^(0.52)(1 − 0.52)/(0.05)^2^ = 383.55.

We anticipated a response rate of about 80% and increased the sample size to 500 community health centres. If one-third of physicians and nurses (*n* = 6) from each centre responded to the survey, it would result in over 2300 individual participants. Such a sample size would exceed the minimal sample size requirement for multivariate regression analyses involving 14 independent variables in this study [26].

A multi-stage stratified sampling strategy was adopted to select survey participants (Figure 1). At the first stage, two municipalities from each of the 26 provinces were identified: the capital city and the city with the third-highest GDP in its respective province. At the second stage, 500 community health centres were identified from the 52 provincial-overseen municipalities and the 4 central-overseen municipalities: 8 to 10 community health centres with an equal geographic distance in each municipality. An invitation letter was sent to the selected centres, asking for support from the senior managers. Each was asked to report the adoption of EMRs in the centre and invite six health workers (physicians and nurses registered with the local health authorities) to participate in the online survey. It was up to the senior managers to decide the ratio between physician and nurse respondents. This resulted in a total of 2719 valid respondents from 462 community health centres. Of the returned questionnaires, 2213 were submitted by physicians and 506 were from nurses. They represented about 19.6% of physicians and 4.4% of nurses employed by the participating community health centres.

### 2.3. Data Collection

The survey measured perceptions of primary care physicians and nurses on the value of EMRs, including their preferred features. This enabled all of the respondents, regardless of whether their centres adopted EMRs, to express their views on the potential benefits patients could gain from the effective support of EMRs. All of the primary care physicians and nurses received EMR training in their qualification courses. This study assessed the perceived value of EMRs in supporting clinical decisions, information exchange, health inquiries, and personalised care. According to Huang et al. [3], these value metrics fall into the categories of complex organisational and system integration functions.

For the community health centres with existing EMRs, the alignment of the existing EMRs with the SOAP (Subjective, Objective, Assessment, Plan) note requirements, a widely adopted structure that provides a cognitive framework for reasoning activities in primary care [27], was also assessed. In addition, the functionality of the existing EMRs was assessed in terms of its support to referral arrangements and telehealth services (defined as the use of telecommunications technology to provide services remotely) and its ability to capture (comprehensive) data that could meet all of the government-defined task needs of community health services.

We consulted six government officials, health managers, and community health workers about the appropriateness of the questionnaire instrument. This included a visit to five community health centres in Beijing and Shanghai. Revisions to the survey instrument were made accordingly in line with the advice received.

The final version of the questionnaire comprised four parts to collect data with regard to (1) sociodemographic characteristics of respondents, such as gender, age, qualification, and job position; (2) support of the existing EMR systems to SOAP note requirements and functionality (measuring support for the comprehensiveness, continuity and coordination of care); (3) perceptions of the value of EMRs in relation to the efficiency and effectiveness of care processes, and; (4) importance ranking—respondents were asked to identify one EMR feature they believed to be the most important in supporting the essential primary care tasks as advocated by academics in the context of the Chinese system [28], which included appointment scheduling, referral arrangements, postoperative follow-up and rehabilitation, management of chronic conditions, and treatment of common diseases. The respondents were also asked to select one or more preferred usability features from a checklist. Usability was defined as how easy and effective an EMR can be used for supporting the day-to-day tasks of a user [29,30,31], which included “easy to use”, “structured documentation”, “quality presentation”, “fast records”, “data restoration”, “fast response”, “error correction”, and “compatibility with multiple devices”.

The questionnaire was uploaded to the Tencent platform. A letter explaining the purpose and protocol of the study, along with a link to the survey, was sent to the senior executives of the participating community health centres by the local health commissions as advised by the National Health Commission. The senior executives of the participating health centres then invited their physicians or nurses to complete the online survey.

The survey was open from April to June 2018. The survey was voluntary. Each survey took on average 15 min to complete. Only one attempt was allowed. Participants were advised to read the informed consent letter before proceeding to the questionnaire and that submission of the online survey was deemed informed consent. A total of 21 questionnaires which violated the pre-defined logic or contained incomplete data were excluded from data analyses (Figure 1).

### 2.4. Statistical Analysis

The functionality of the existing EMRs was measured on a dichotomous scale (1 = adopted, 0 = not adopted) at the organisational level. Of the 271 (58.7%) community health centres with an EMR system, the percentage of available functions was calculated and compared across the eastern, central and western regions using Pearson (crosstab) Chi-square tests.

The perceptions of individual respondents (*n* = 2719) on the value of EMRs were rated on a five-point Likert scale (1 = strongly disagree, 2 = disagree, 3 = slightly agree, 4 = agree, 5 = strongly agree), with a higher score indicating a higher perceived value. Rank sum tests were performed to examine differences in the perceptions across the eastern, western and central respondents and between those with and without EMRs. Multivariate ordinal regression models with an enter approach were established to determine the associations between the sociodemographic (independent) variables and the perceived value of EMRs (dependent variables) adjusting for variations in other variables.

The perceived importance of EMR features was estimated by the proportional distribution of voting from the respondents. The frequency of each of the preferred usability features cited by the respondents was ranked in order. Pearson (crosstab) Chi-square tests were performed to test the regional differences and the differences between those with and without EMRs in the voting distribution of these two indicators.

Data were extracted into an Excel spreadsheet (Microsoft, Redmond, WA, USA) and analysed with SPSS 22.0 (IBM SPSS, Armonk, NY, USA). A *p*-value less than 0.05 was deemed statistically significant. Some of the variables contained a small number of cases (ranging from 1 to 3) with missing values. A pairwise deletion method was used for managing the missing data.

## 3. Results

### 3.1. Characteristics of Respondents

In total, 2719 questionnaires were returned. A slightly higher proportion (39%) of respondents came from the central region (1066), compared to the other two regions (847 from the eastern and 806 from the western). Just over half (54%) worked in an urban setting. The respondents’ average age was 38.1 years (Standard Deviation = 8.6) and 39.6% were male. Most were physicians (81.4%) and engaged in clinical services (76.5%). The majority (57%) had a bachelor’s degree as their highest qualification and 51.7% held a junior professional title (Table 1).

Significant regional differences in the socio-demographic characteristics of the respondents were found, although there was no difference in their experience of using EMRs (Table 1). The respondents who were male, younger, worked as a doctor in an urban setting, had a tertiary degree and higher professional title, and engaged in more clinical services were more likely to reside in the more developed eastern region and use EMRs.

### 3.2. Coverage and Functionality of EMRs

The majority (58.7%) of the participating community health centres reported the existence of an EMR system. No differences in the adoption of EMRs were found across the three regions (χ^2^ = 2.475, *p* = 0.290) and between urban and rural (χ^2^ = 2.227, *p* = 0.081). Of the EMRs installed in the 271 community health centres, 70.5% met the SOAP note requirements. However, less than half (38.4%) captured data that were perceived as comprehensive (be able to meet all task needs). Half of the EMR systems had access to telehealth support for laboratory, imaging and patient consultation services. Only 35.4% supported referral arrangements. The EMR systems in the relatively more developed eastern region were more likely to capture data that met all of the task needs and support patient referral arrangements (*p* < 0.05) compared with their less-developed western and central counterparts (Table 2).

### 3.3. Perceived Value of EMRs

Most respondents (89.4%) considered that it was necessary to establish an EMR system in primary care. The EMR functions endorsed (“agree” or “strongly agree”) by the respondents covered all aspects evaluated, including decision support for diagnosis and treatment (76.1%), sharing of information across organisations (77.7%), higher work efficiency (76.6%), and real-time health inquiries (78.9%). Respondents from the least developed western region were less likely (*p* < 0.05) to appreciate the value of EMRs compared with those in the more developed eastern and central regions. The ratings between those with and without EMRs were similar, except for one aspect—where those with EMRs were more likely to recognise the value of EMRs in “facilitating patient contracts with general practitioners” (Table 3).

The regional differences in the perceived value of EMRs were still statistically significant after adjustment for variations in the sociodemographic characteristics of the respondents, except for “facilitating patient contracts with general practitioners” (Table 4). Rural respondents were less likely to appreciate the value of EMRs in facilitating health contracts than their urban counterparts. Respondents with a clinical job were less likely to recognise that EMRs would improve work efficiency and support real-time enquiries than others. The absence of EMRs in the CHCs was associated with lower levels of the perceived value of EMRs after adjustments for variations in other variables (Table 4).

### 3.4. Preferred Functional Features of EMRs

About 66% of the respondents voted “management of chronic conditions” as the most important feature of EMRs, followed by 14% for referral arrangements and 13% for appointment scheduling. Only 5% endorsed “treatment of common diseases” as the most important feature of EMRs for primary care (Figure 2). No significant difference was found in the ranking across the eastern, central and western respondents (χ^2^ = 14.475, *p* = 0.070). The ranking of the functional features remained unchanged regardless of the existence of EMRs. However, the respondents with EMRs were slightly less likely to choose “management of chronic conditions” as the most important feature compared with those without EMRs (63% vs. 71%, *p* < 0.001).

### 3.5. Preferred Usability Features of EMRs

“Easy to use” was the most cited usability feature (90.4%). This was followed by “structured documentation” (88.8%). “Fast response”, “error correction” and “compatibility with multiple devices” were least likely to be cited, with less than 80% of respondents ticking these boxes. The order of ranking was consistent across regions and between those with and without EMRs. The western respondents were less likely to consider “fast records” as a preferred usability feature compared with their eastern and central counterparts. Higher levels of endorsement of “easy to use”, “structured documentation”, “fast response” and “compatibility of multiple devices” were found in those with EMRs compared with those without EMRs (Table 5). 

## 4. Discussion

This study found that EMR adoption in community health services has failed to meet the central target across all regions in mainland China. Although in general there is a high level of recognition of the value of EMRs from the primary care physicians and nurses, significant regional differences exist. Those from the relatively more developed eastern region are more likely to appreciate the value of EMRs compared with their counterparts from other regions. There is a consistent pattern in preferred usability features across regions and between those with and without EMRs, with “easy to use” and “structured documentation” being ranked on the top of the list.

### 4.1. Adoption of EMRs in Primary Care

Overall, the computerisation of medical records in primary care is gaining momentum in China: more than 58% of community health centres have established EMRs. In the most recent round of health reform, the development of health information technologies is considered as one of the essential building blocks for the modernisation of the health system in China. However, relatively slow progress in the less developed regions is anticipated due to restrictions in financial and human resources [32]. In China, health financing responsibilities have been devolved and great disparities exist in financial capacities across regions [33]. As a result, central subsidies were given to the less developed central and western regions to balance the development. From 2011 to 2015, 3 billion yuan (US$ 454 million) were injected by the central government to support the adoption of EMRs in primary care in the central and western regions [34]. Substantial governmental investments are often required to achieve universal adoption of EMRs as is evidenced in the developed countries [35]. This usually includes governmental certification and incentive programs to push for meaningful use of EMRs [36]. From 2001 to 2015, for example, the Canadian government spent $2 billion in EMR development, which resulted in a rapid increase in the percentage of Canadian physicians using EMRs: rising from 25% in 2007 to 75% in 2014 [20,37]. Similarly, the percentage of federally qualified health centres with an EMR system in the US increased from 40% in 2009 to 93% in 2013 [38]. The universal adoption of EMRs in primary care was only achieved in the mid-2000s in some developed countries, such as the UK, Netherlands, Australia and New Zealand [39,40], although some developed countries such as Canada and the US have only started to catch up with their wealthy counterparts in recent years [13,41].

Interestingly, no regional difference in the adoption of EMRs was found in community health services across regions in China, nor between urban and rural facilities. Unbalanced development in EMRs was observed in the US [42] as rural health facilities lagged far behind their urban counterparts. The absence of EMRs in some community health services in China is likely to be associated with the governmental priority in getting community health services widely covered. Community health services started to gain momentum in China in the late 1990s. Universal coverage has been the priority often at the cost of poor quality. Community health services are highly dependent on local governmental investment. Unlike hospitals, they do not have their own financial capacity to fund the development of health information systems. It is important to note that this study did not cover the large number of small clinics, where EMRs barely exist. According to the 2019 health statistics of China [43], there are 857,087 small clinics in China, contributing to 28.7% of outpatient care services.

The low level of the recognition of “meaningful use” of the existing EMRs in primary care is more concerning in China. Only 50% of the existing EMRs in community health services offered telehealth support to laboratory, imaging and patient consultation services. This is in sharp contrast with the 76.1% endorsement (“agree” or “strongly agree”) from the respondents for EMRs to support decision making on diagnosis and treatment. The use of EMRs for referral arrangements is even lower in community health services at a level of 35.4%. In Australia, 85% of general practitioners reported using EMRs to order laboratory tests and 84% generated health summaries from EMRs that could be used for patient referral [40].

Although there is no regional disparity in the overall computerisation of medical records in community health services in China, the EMR systems in the most developed eastern region are more likely to capture data that meet all of the task needs and support patient referral arrangements in particular compared with their less-developed western and central counterparts. The unbalanced socioeconomic development across regions in China may have a great impact on the allocation of health resources [44]. The community health services located in the relatively more developed regions are likely to attract higher levels of governmental investment, enabling them to adopt an EMR with more functional modules. This is not unique to China. In the US, for example, the prioritisation of the meaningful use of EMRs in resource-poor facilities is also advocated [45]. With increasing population mobility in the modern world, it is expected that patients would be increasingly attracted to resource-rich facilities and seek medical care outside their residential region. Therefore, data should follow the patient instead of being confined within the boundary of the health facility [14].

### 4.2. Value of EMRs as Perceived by Primary Care Workers

There was overwhelming agreement among the respondents of this study on the value of EMRs. Almost 90% of the primary care workers believed that it was necessary to implement EMRs in community health services. All six aspects of EMRs value evaluated in this study were appreciated (“agree” or “strongly agree”) by at least 70% of the respondents. The positive attitudes of primary care workers toward EMRs are also common in other countries [46,47]. Previous studies [48] indicate that perceived usefulness is a key factor affecting EMR use in health workers.

This study found that primary care workers in the socioeconomically disadvantaged (western) region were less likely to appreciate the value of EMRs compared with their counterparts in other regions. This is likely to have contributed to the lower willingness of the health workers from the disadvantaged region to use information technologies [49]. Several studies argue that variations in the attitudes of health workers toward EMRs can be a result of the uneven establishment of EMRs [7,36,50,51,52,53]. Indeed, our ordinal regression analyses indicate that the establishment of EMRs does have a significant positive association with the perceived value of EMRs.

The study shows that EMR functions supporting continuity and coordination of care are most valued by primary care workers. “Management of chronic conditions” was singled out as the most important feature of EMRs, which requires long-term data tracking and data sharing [31]. In contrast, EMR support for the treatment of common diseases was ranked at the bottom. This may simply be a reflection of the nature of information needs. Community health workers in China are often inundated with large workloads in population-based services such as health education, screening and management of chronic conditions. The scope of individual-based clinical care is relatively limited [54].

Empirical evidence in China shows that EMRs can facilitate community management of chronic conditions and improve patient care outcomes [55,56]. However, there is a lack of assessment on the overall impact of EMRs in primary care [32]. In some developed countries, the adoption of EMRs is expected to deliver significant savings (for example an annual saving of US$ 80 billion for the US health system) through increased efficiency and quality of health care services [57].

### 4.3. Preferred Usability Features of EMRs in Primary Care Workers

It is important to note that health workers do not always take advantage of all the EMR functions even if they are available. Several studies in Canada show that Canadian doctors used 65% of all EMR functions on average [58], fewer than 80% exclusively used EMRs to record patient care data [20], and 44% reported poor compatibility with other electronic systems [59]. A mismatch between EMR functionality and the needs of primary care workers is not uncommon [35].

Poor usability could potentially jeopardise the implementation and use of EMRs in primary care. This is evidenced by the wide range of preferred usability features cited by the participants in this study. Of the cited usability features, “easy to use” and “structured documentation” were at the top of the list. If an EMR system is too complicated and difficult to use, its perceived usefulness will deteriorate. Training is often needed over a long period of time as an integral part of the implementation of EMRs [60]. From a cognitive perspective, “structured documentation” is required to ease the readability and comprehension of the data available in EMRs [61].

Although “compatibility with multiple devices” was ranked relatively low by the primary care workers in this study, such a perception may change over time when the EMRs become more mature. In the US, primary care organisations are relatively satisfied with the data acquisition, documentation, and result tracking functions of their EMRs [62], but there are increasing concerns about information and system interoperability [20,63]. Some countries have started to develop an electronic health records (EHR) system, which is patient-oriented and collects aggregated and longitudinal data about a patient over a wide area network [14].

## 5. Strengths and Limitations

To our knowledge, this is the first study of its kind on a nationwide sample. The study had a high response rate and a large sample size. However, the cross-sectional design prevented us from drawing causal conclusions. The study focused on the EMR functions supporting clinical practices in community health services, which did not include those that support research and public health services. The sample may be biased as the vast number of small clinics were not included and the selection of participants was not randomised. Further studies are needed to explore how primary care workers use EMRs to support decision making and how EMRs can serve for the purpose of research and public health services such as the early detection and warning of emerging infectious diseases [64].

## 6. Conclusions

In summary, more than 58% of community health services have adopted EMRs in mainland China. However, the limited recognition of the meaningful use of established EMRs is concerning. The low level of functionality in data acquisition and referral arrangements runs counter to the requirements for “management of chronic conditions”, the most preferred functional feature of EMRs in community health services. Regional disparities in the realised value of EMRs in community health services also deserve increasing policy attention.

Strategies for strengthening EMRs should emphasise the usefulness of EMRs in supporting the clinical tasks of primary care workers [65]. Financial subsidies and incentive policies from governments are also important for the wide and equitable adoption of EMRs in primary care facilities [66,67]. In a large country such as China where imbalanced socioeconomic development is evident, central financial subsidies to the less developed regions are usually critical. Meanwhile, local governments that are delegated with the responsibility of local health development should prioritise primary care as it is the most important element in efforts to achieve health equality.

The findings of this study also have some implications for the further development of health information standards in China. Information exchange and interoperability have emerged as an important consideration in the development of EMRs. They will become increasingly important when the role of primary care workers is further enhanced for the coordinated continuous care in relation to chronic conditions. While these functional features were not kept in the essential requirements in the current health information standards for concerns of financial affordability in the less developed regions, it is important to maintain a regular update of the standards. Apart from the financial support, technical support to the local governments and health institutions in the less developed regions is equally important. At the same time, support for primary care workers to be trained in the effective use of EMRs has to be strengthened. Despite the rapid adoption of information technologies, there is a serious shortage of skilled workforce in this area [32]. Medical informatics has only recently become a genuine discipline in tertiary education in China. There is an urgent need to better align the adoption of EMRs with the information needs of health care workers and consumers.

## Figures and Tables

**Figure 1 ijerph-17-08510-f001:**
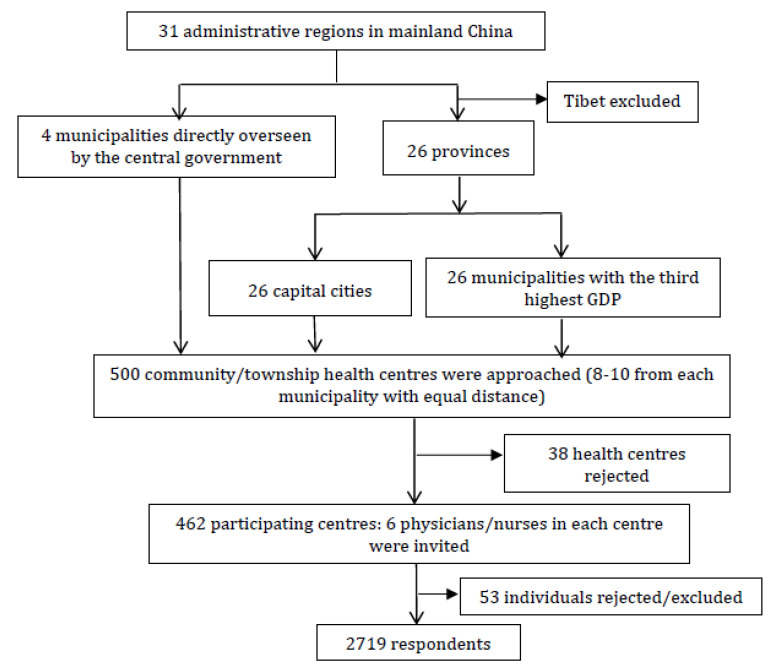
Flow chart of the sampling procedure.

**Figure 2 ijerph-17-08510-f002:**
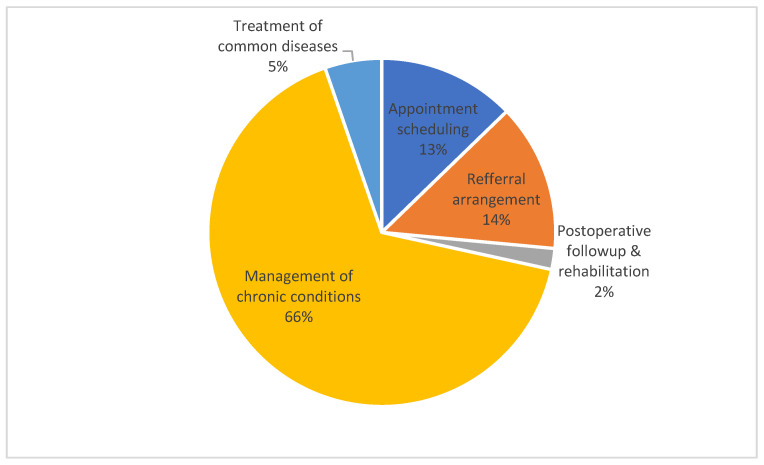
Preferred functional features ranked by respondents.

**Table 1 ijerph-17-08510-t001:** Socio-demographic characteristics of respondents.

Characteristics	Sample Size	Number (Percentage) of Respondents				Number (Percentage) of Respondents		
	N (%)	Eastern	Central	Western	*p*	With EMRs	Without EMRs	*p*
**Gender**					<0.001			<0.001
Male	1076 (39.6%)	356 (42.0%)	442 (41.5%)	278 (34.5%)		691 (42.9%)	385 (34.7%)	
Female	1643 (60.4%)	491 (58.0%)	624 (58.5%)	528 (65.5%)		919 (57.1%)	724 (65.3%)	
**Age (years)**					0.014			0.006
18–44	2013 (74.0%)	651 (76.9%)	801 (75.1%)	561 (69.5%)		1204 (74.8%)	809 (72.9%)	
45–59	668 (24.6%)	186 (22.0%)	251 (23.5%)	231 (28.7%)		393 (24.4%)	275 (24.8%)	
≥60	38 (1.4%)	10 (1.1%)	14 (1.4%)	14 (1.8%)		13 (0.8%)	25 (2.3%)	
**Profession**					0.001			<0.001
Physician	2213 (81.4%)	714 (84.3%)	876 (82.2%)	623 (77.3%)		1368 (85.0%)	845 (76.2%)	
Nurse	506 (19.6%)	133 (15.7%)	190 (17.8%)	183 (22.7%)		242 (15.0%)	264 (23.8%)	
**Job**					<0.001			<0.001
Clinical service	2080 (76.5%)	689 (81.3%)	820 (76.9%)	571 (70.8%)		1296 (80.5%)	784 (70.7%)	
Technical support	105 (3.8%)	37 (4.4%)	35 (3.3%)	33 (4.1%)		43 (2.7%)	62 (5.6%)	
Preventive care	219 (8.1%)	53 (6.3%)	96 (9.0%)	70 (8.7%)		111 (6.9%)	108 (9.7%)	
Others	315 (11.6%)	68 (8.0%)	115 (10.8%)	132 (16.4%)		160 (9.9%)	155 (14.0%)	
**Qualification**					<0.001			<0.001
Associate degree	1083 (39.8%)	267 (31.5%)	437 (41.0%)	379 (47.0%)		571 (35.5%)	511 (46.1%)	
Bachelor degree	1550 (57.0%)	537 (63.4%)	595 (55.8%)	418 (51.9%)		974 (60.6%)	576 (51.9%)	
Postgraduate degree	86 (3.2%)	43 (5.1%)	34 (3.2%)	7 (1.1%)		62 (3.9%)	22 (2.0%)	
**Title**					<0.001			0.030
Junior	1405 (51.7%)	382 (45.1%)	552 (51.8%)	471 (58.4%)		797 (49.5%)	608 (54.8%)	
Middle	992 (36.5%)	358 (42.3%)	381 (35.7%)	253 (31.4%)		607 (37.7%)	385 (34.7%)	
Associate senior	291 (10.7%)	96 (11.3%)	120 (11.3%)	75 (9.3%)		184 (11.4%)	107 (9.6%)	
Senior	31 (1.1%)	11 (1.3%)	13 (1.2%)	7 (0.9%)		22 (1.4%)	9 (0.8%)	
**Location**					<0.001			0.003
Urban	1475 (54.3%)	503 (59.4%)	524 (49.2%)	448 (55.6%)		909 (56.5%)	566 (51.0%)	
Rural	1243 (45.7%)	334 (40.6%)	542 (50.8%)	357 (44.4%)		700 (43.5%)	543 (49.0%)	
**Use of EMRs**					0.201			
Yes	1610 (59.2%)	481 (56.8%)	638 (59.8%)	491 (60.9%)		-	-	-
No	1109 (40.8%)	366 (43.2%)	428 (40.2%)	315 (39.1%)		-	-	-

Note: EMRs—Electronic Medical Records.

**Table 2 ijerph-17-08510-t002:** Functionality of electronic medical records adopted by the community/township health centres (*n* = 271).

Functionality	Number (Percentage) of Health Centres				χ^2^	*p*
	**Total**	**Eastern**	**Central**	**Western**		
**Aligned with SOAP note requirements ***					2.461	0.292
Yes	191 (70.5%)	63 (66.3%)	63 (76.8%)	65 (69.1%)		
No	80 (29.5%)	32 (33.7%)	19 (23.2%)	29 (30.9%)		
**Access to telehealth support for laboratory, imaging, or patient consultation services**					3.622	0.163
Yes	136 (50.2%)	52 (54.7%)	34 (41.5%)	50 (53.2%)		
No	135 (49.8%)	43 (45.3%)	48 (58.5%)	44 (46.8%)		
**Referral arrangements**					11.385	0.003
Yes	96 (35.4%)	45 (47.4%)	19 (23.2%)	32 (35.4%)		
No	175 (64.6%)	50 (52.6%)	63 (76.8%)	62 (64.6%)		
**Captured data that meet all job needs**					7.768	0.021
Yes	104 (38.4%)	46 (48.4%)	31 (37.8%)	27 (28.7%)		
No	167 (61.6%)	49 (51.6%)	51 (62.2%)	67 (71.3%)		

Note: * SOAP indicates subjective, objective, assess and plan.

**Table 3 ijerph-17-08510-t003:** Perceived value of electronic medical records in primary care (*n* = 2719).

Value in Supporting	Number (Percentage) of Respondents					H	*p*
	Strongly Disagree	Disagree	Slightly Agree	Agree	Strongly Agree		
**Decision making in diagnosis and treatment**							
Eastern	83 (9.8%)	9 (1.1%)	94 (11.1%)	336 (39.7%)	325 (38.4%)	12.29	<0.01
Central	113 (10.6%)	10 (0.9%)	105 (9.8%)	460 (43.2%)	378 (35.5%)		
Western	108 (13.4%)	25 (3.1%)	103 (12.8%)	303 (37.6%)	267 (33.1%)		
With EMRs	184(11.6%)	18 (1.1%)	173 (10.9%)	651 (41.1%)	558 (35.2%)	6.85	0.14
Without EMRs	120(10.6%)	26 (2.3%)	129 (11.4%)	448 (39.5%)	412 (36.3%)		
**Information exchange across organisations**							
Eastern	83 (9.8%)	21 (2.5%)	71 (8.4%)	336 (39.7%)	336 (39.7%)	7.56	0.02
Central	114 (10.7%)	16 (1.5%)	88 (8.3%)	436 (40.9%)	412 (38.6%)		
Western	103 (12.8%)	20 (2.5%)	91 (11.3%)	304 (37.7%)	288 (35.7%)		
With EMRs	184 (11.6%)	26 (1.6%)	144 (9.1%)	631 (39.8%)	599 (37.8%)	5.11	0.28
Without EMRs	116 (10.2%)	31 (2.7%)	106 (9.3%)	445 (39.2%)	437 (38.5%)		
**Improving work efficiency**							
Eastern	83 (9.8%)	16 (1.9%)	84 (9.9%)	315 (37.2%)	349 (41.2%)	11.40	<0.01
Central	111 (10.4%)	20 (1.9%)	93 (8.7%)	431 (40.4%)	411 (38.6%)		
Western	108 (13.4%)	24 (3.0%)	96 (11.9%)	291 (36.1%)	287 (35.6%)		
With EMRs	122(10.7%)	27 (2.4%)	119 (10.5%)	429 (37.8%)	438 (38.6%)	0.93	0.92
Without EMRs	180(11.4%)	30 (1.9%)	154 (9.7%)	608 (38.4%)	609 (38.4%)		
**Personalised care**							
Eastern	79 (9.3%)	21 (2.5%)	93 (11.0%)	335 (39.6%)	319 (37.7%)	12.91	<0.01
Central	108 (10.1%)	17 (1.6%)	108 (10.1%)	441 (41.4%)	392 (36.8%)		
Western	105 (13.0%)	21 (2.6%)	106 (13.2%)	316 (39.2%)	258 (32.0%)		
With EMRs	114(10.0%)	29 (2.6%)	129 (11.4%)	451 (39.7%)	412 (36.3%)	2.55	0.64
Without EMRs	178(11.2%)	21 (2.5%)	178 (11.2%)	641 (40.5%)	557 (35.2%)		
**Real-time health inquiries**							
Eastern	82 (9.7%)	13 (1.5%)	67 (7.9%)	333 (39.3%)	352 (41.6%)	7.38	0.03
Central	111 (10.4%)	18 (1.7%)	86 (8.1%)	423 (39.7%)	428 (40.2%)		
Western	106 (13.2%)	17 (2.1%)	73 (9.1%)	312 (38.7%)	298 (37.0%)		
With EMRs	120 (10.6%)	19 (1.7%)	96 (8.5%)	451 (39.7%)	449 (39.6%)	0.57	0.97
Without EMRs	179 (11.3%)	29 (1.8%)	130 (8.2%)	617 (39.0%)	629 (39.7%)		
**Facilitating patient contracts with general practitioners**							
Eastern	82 (9.7%)	14 (1.7%)	92 (10.9%)	311 (36.7%)	348 (41.1%)	9.58	0.01
Central	111 (10.4%)	19 (1.8%)	121 (11.4%)	413 (38.7%)	402 (37.7%)		
Western	107 (13.3%)	17 (2.1%)	103 (12.8%)	294 (36.5%)	285 (35.4%)		
With EMRs	188 (11.7%)	35 (2.2%)	157 (9.8%)	565 (35.1%)	665 (41.3%)	32.46	0.00
Without EMRs	112 (10.1%)	15 (1.4%)	159 (14.3%)	453 (40.8%)	370 (33.4%)		

Note: EMRs—Electronic Medical Records.

**Table 4 ijerph-17-08510-t004:** Factors associated with the perceived value of electronic medical records: results of ordinal logistic regression models (*n* = 2719).

	Decision making in Diagnosis and Treatment			Supporting Share of Information			Improving Work Efficiency			Personalised Care			Supporting Real-Time Enquiries			Facilitating Health Contracts		
	Coefficient (95% CI)			Coefficient (95% CI)			Coefficient (95% CI)			Coefficient (95% CI)			Coefficient (95% CI)			Coefficient (95% CI)		
**Region (Reference = Western)**																		
Eastern	0.317 **	(0.135	0.499)	0.241 **	(0.058	0.423)	0.315 **	(0.133	0.498)	0.316 **	(0.134	0.498)	0.262 **	(0.079	0.446)	0.069	(−0.114	0.251)
Middle	0.254 **	(0.083	0.424)	0.230 **	(0.059	0.401)	0.257 **	(0.086	0.427)	0.306 **	(0.136	0.477)	0.206 *	(0.035	0.378)	−0.091	(−0.262	0.080)
**Rural (vs. Urban)**	−0.073	(−0.219	0.072)	−0.145	(−0.291	0.000)	−0.067	(−0.212	0.078)	−0.031	(−0.176	0.114)	−0.098	(−0.244	0.048)	−0.668 **	(−0.815	−0.522)
**Female (vs. Male)**	0.042	(−0.116	0.199)	0.028	(−0.130	0.185)	0.027	(−0.130	0.184)	0.115	(−0.042	0.272)	0.031	(−0.127	0.190)	0.056	(−0.101	0.213)
**≥45 years age (vs. <45)**	0.063	(−0.126	0.252)	0.101	(−0.089	0.291)	0.118	(−0.071	0.308)	0.137	(−0.052	0.326)	0.105	(−0.086	0.295)	0.016	(−0.173	0.206)
**Nurse (vs. physician)**	0.048	(−0.158	0.254)	0.128	(−0.079	0.335)	0.075	(−0.131	0.281)	0.024	(−0.182	0.229)	0.045	(−0.162	0.252)	0.009	(−0.198	0.215)
**Job (reference = Others)**																		
Clinical	−0.191	(−0.422	0.040)	−0.146	(−0.377	0.085)	−0.235 *	(−0.466	−0.004)	−0.172	(−0.402	0.059)	−0.234 *	(−0.467	−0.001)	−0.168	(−0.401	0.064)
Technician	0.058	(−0.354	0.470)	0.296	(−0.121	0.713)	0.083	(−0.331	0.497)	0.108	(−0.304	0.521)	0.017	(−0.399	0.432)	−0.325	(−0.735	0.086)
Preventive care	0.030	(−0.290	0.351)	0.111	(−0.211	0.433)	0.014	(−0.308	0.335)	0.064	(−0.257	0.385)	0.044	(−0.280	0.368)	−0.150	(−0.472	0.172)
**Qualification (Reference = Postgraduate)**																		
Associate degree	−0.193	(−0.624	0.238)	−0.204	(−0.636	0.228)	−0.163	(−0.593	0.266)	−0.114	(−0.542	0.314)	−0.081	(−0.513	0.350)	0.281	(−0.147	0.709)
Bachelor degree	0.011	(−0.399	0.420)	0.038	(−0.373	0.448)	0.098	(−0.310	0.506)	0.140	(−0.267	0.547)	0.139	(−0.271	0.549)	0.297	(−0.109	0.703)
**Title (Reference = Senior)**																		
Junior	−0.042	(−0.301	0.217)	−0.002	(−0.261	0.258)	0.024	(−0.235	0.283)	0.170	(−0.088	0.428)	0.002	(−0.258	0.263)	0.116	(−0.143	0.375)
Middle	0.050	(−0.194	0.295)	0.021	(−0.224	0.266)	0.034	(−0.211	0.278)	0.122	(−0.121	0.365)	0.026	(−0.220	0.272)	0.025	(−0.219	0.269)
**No EMRs (vs with EMRs)**	−0.296 **	(−0.440	−0.152)	−0.241 **	(−0.385	−0.097)	−0.262 **	(−0.406	−0.119)	−0.225 **	(−0.369	−0.082)	−0.283 **	(−0.427	−0.138)	−0.002	(−0.146	0.142)

Note: 95% CI = 95% Confidence Interval; * *p* < 0.05; ** *p* < 0.01.

**Table 5 ijerph-17-08510-t005:** Preferred features of usability of electronic medical records (*n* = 2719).

Features	Total			Eastern			Central			Western			χ^2^	*p*	With EMRs			Without EMRs			χ^2^	*p*
	Order	n	(%)	Order	n	(%)	Order	n	(%)	Order	n	(%)			Order	n	(%)	Order	n	(%)		
**Easy to use**	1	2459	(90.4)	1	780	(92.1)	1	963	(90.3)	1	716	(88.8)	5.083	0.079	**1**	**1473**	**(93.0)**	**2**	**986**	**(86.9)**	**28.639**	**<0.001**
**Standardised documentation**	2	2419	(89.0)	2	752	(88.8)	1	963	(90.3)	2	704	(87.3)	4.230	0.121	**2**	**1427**	**(90.1)**	**1**	**992**	**(87.4)**	**4.865**	**0.030**
**Quality records**	3	2298	(84.5)	3	720	(85.0)	4	898	(84.2)	3	680	(84.3)	0.231	0.891	4	1337	(84.4)	3	961	(84.7)	0.035	0.872
**Fast records**	**4**	**2284**	**(84.0)**	**4**	**710**	**(83.8)**	**3**	**921**	**(86.4)**	**4**	**653**	**(81.0)**	**9.915**	**0.007**	3	1349	(85.2)	4	935	(82.4)	3.817	0.056
**Security of data**	5	2197	(80.8)	5	672	(79.3)	5	872	(81.8)	4	653	(81.0)	1.879	0.391	5	1289	(81.4)	5	908	(80.0)	0.807	0.375
**Fast response**	6	1989	(73.2)	6	634	(74.9)	8	763	(71.6)	6	592	(73.4)	2.631	0.268	**6**	**1230**	**(77.7)**	**8**	**759**	**(66.9)**	**39.118**	**<0.001**
**Low occurrence of errors**	7	1979	(72.8)	7	621	(73.1)	7	766	(71.9)	6	592	(73.4)	0.764	0.683	8	1164	(73.5)	6	815	(71.8)	0.941	0.337
**Interoperability of multiple devices**	8	1971	(72.5)	8	600	(70.8)	6	790	(74.1)	8	581	(72.1)	2.626	0.629	**7**	**1193**	**(75.3)**	**7**	**778**	**(68.5)**	**15.194**	**<0.001**

Note: Figures in bold indicate statistical significance with *p* < 0.05.

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
