# Peer review of "Perceived Value of Electronic Medical Records in Community Health Services: A National Cross-Sectional Survey of Primary Care Workers in Mainland China"

_ijerph, 2020, doi:10.3390/ijerph17228510_

Round 1
Reviewer 1 Report
Thank you for a very interesting manuscript addressing an important aspect of research. A couple of minor comments:
While you state the rate of adoption of EMR in primary care in China over the years and different regions, the impact of doing so is less well-defined. Perhaps a quantification (in monetary value or work hours saved, etc.) -if that information exists- could be indicative of the magnitude of the task. It is useful to have a %, however it does not always reveal the scale.
It would be interesting to know (if that information exists) in how many of the less developed areas of China the development of EMR in primary healthcare is a priority as announced by the local authorities themselves. It might demonstrate an understanding of the added-value by the decision-makers, even if it does not materialise in practice.
You mention technical support and training very briefly in the last two sentences of the conclusion. These are important aspects that have proven critical to EMR adoption in other settings. Therefore each one would need to be expanded with 1-2 short sentences and an additional 1-2 references supporting your statements.
Author Response
Reviewer 1:
Thank you for a very interesting manuscript addressing an important aspect of research. A couple of minor comments:
While you state the rate of adoption of EMR in primary care in China over the years and different regions, the impact of doing so is less well-defined. Perhaps a quantification (in monetary value or work hours saved, etc.) -if that information exists- could be indicative of the magnitude of the task. It is useful to have a %, however it does not always reveal the scale.
Response: Thanks for the constructive advice. We have added some discussions in the discussion section in regard to government investments in EMRs in primary care and its impacts on primary care services in China.
Despite a unified top-down approach in primary care development in China, local governments were delegated with responsibilities to invest in facility infrastructures including health information systems. However, central subsidies were given to the less developed middle and western regions to balance the development. From 2011 to 2015, 3 billion yuan (US$ 454 million) were injected by the central government to support the adoption of EMRs in primary care in the middle and western regions [1]. (Line 309-317 on page 12 in clean version)
Empirical evidence in China shows that EMRs can facilitate community management of chronic conditions and improve patient care outcomes [2,3]. But there is a lack of assessment on the overall impact of EMRs in primary care [4]. In some developed countries, the adoption of EMRs is expected to deliver significant savings (for example an annual saving of US$ 80 billion in the US system) through increased efficiency and quality of health care services [5]. [1]. (Line 381-385 on page 13 in clean version)
It would be interesting to know (if that information exists) in how many of the less developed areas of China the development of EMR in primary healthcare is a priority as announced by the local authorities themselves. It might demonstrate an understanding of the added-value by the decision-makers, even if it does not materialise in practice.
Response: Thanks. We have added some discussions in line with the advice.
In the most recent round of health reform, the development of health information technologies is considered as one of the essential building blocks for modernisation of the health system in China. However, relatively slow progress in the less developed regions is anticipated due to restrictions in financial and human resources [4]. In China, health financing responsibilities have been devolved and there exist great disparities in financial capacities across regions [6]. As a result, central subsidies were given to the less developed middle and western regions to balance the development. [1]. (Line 309-317 on page 12 in clean version)
You mention technical support and training very briefly in the last two sentences of the conclusion. These are important aspects that have proven critical to EMR adoption in other settings. Therefore each one would need to be expanded with 1-2 short sentences and an additional 1-2 references supporting your statements
Response: Thanks. We have added some discussions in line with the advice.
Despite a rapid adoption of information technologies, there is a serious shortage of skilled workforce in this area [4]. Medical informatics has only recently become a genuine discipline in tertiary education in China. There is an urgent need to better align the adoption of EMRs with the information needs of health care workers and consumers. (Line 439-442page 14 in clean version)
Reference
- Yuefeng Li; Jianping Hu; Qun Meng. SWOT analysis on population health information development in China Chinese Journal of Health Information Management 2016, 13, 45-50 DOI: 10.3969/j.issn.1672-5166.2016.3901.3909.
- Li, D.; Wei, X.; Wu, H.; Liu, X.; Ge, C.; Gao, W. Effect of an intelligent family physician-optimised coordination model program on hypertension management in a Beijing community. Aust J Prim Health 2020, 26, 402-409, doi:10.1071/py19218.
- Xue-Juan, W.; Hao, W.; Cai-Ying, G.; Xin-Ying, L.; Hong-Yan, J.; Li, W.; Xiao-Ling, G.; Wan-Ying, L.; Wen-Juan, G.; Wan-Nian, L. Impact of an intelligent chronic disease management system on patients with type 2 diabetes mellitus in a Beijing community. BMC Health Serv Res 2018, 18, 821, doi:10.1186/s12913-018-3610-z.
- Lei, J.; Meng, Q.; Li, Y.; Liang, M.; Zheng, K. The evolution of medical informatics in China: A retrospective study and lessons learned. International Journal of Medical Informatics 2016, 92, 8-14, doi:https://doi.org/10.1016/j.ijmedinf.2016.04.011.
- Xue, Y.; Liang, H.; Wu, X.; Gong, H.; Li, B.; Zhang, Y. Effects of electronic medical record in a Chinese hospital: a time series study. Int J Med Inform 2012, 81, 683-689, doi:10.1016/j.ijmedinf.2012.05.017.
- Liu, C.; Legge, D. Challenges in China's health system reform: lessons from other countries. Aust J Prim Health 2017, 23, i-ii, doi:10.1071/PYv23n4_ED.
- O'Donnell, A.; Kaner, E.; Shaw, C.; Haighton, C. Primary care physicians' attitudes to the adoption of electronic medical records: a systematic review and evidence synthesis using the clinical adoption framework. BMC Med Inform Decis Mak 2018, 18, 101, doi:10.1186/s12911-018-0703-x.
- Raymond, L.; Pare, G.; Marchand, M. Extended use of electronic health records by primary care physicians: does the electronic health record artefact matter? Health Informatics J 2017, 25, 71-82, doi:10.1177/1460458217704244.
- Persaud, N. A national electronic health record for primary care. CMAJ 2019, 191, E28-E29, doi:10.1503/cmaj.181647.
- Canadian Institute for Health Information. How Canada Compares: Results From the Commonwealth Fund’s 2019 International Health Policy Survey of Primary Care Physicians; Ottawa, ON: CIHI: 2020.
- Ryan, J.; Doty, M.M.; Abrams, M.K.; Riley, P. The Adoption and Use of Health Information Technology by Community Health Centers, 2009-2013. Issue Brief (Commonwealth Fund) 2014, 1-8.
- Pare, G.; Raymond, L.; Guinea, A.O.; Poba-Nzaou, P.; Trudel, M.C.; Marsan, J.; Micheneau, T. Electronic health record usage behaviors in primary care medical practices: A survey of family physicians in Canada. Int J Med Inform 2015, 84, 857-867, doi:10.1016/j.ijmedinf.2015.07.005.
- Collier, R. National Physician Survey: EMR use at 75%. CMAJ : Canadian Medical Association journal = journal de l'Association medicale canadienne 2015, 187, E17-E18, doi:10.1503/cmaj.109-4957.
Reviewer 2 Report
Thank you for the opportunity to review the manuscript entitled “Perceived value of electronic medical records in community health services: a national cross-sectional survey of primary care workers in mainland China”. In this work, the authors performed an impressive survey on the usage of electronic medical records (EMR) in China. Then, they performed a statistical analysis of the obtained dataset, showing the correlation between a number of factors and availability/usage of EMR. While I acknowledge how valuable is the goal of this study, the persistent narrative of the manuscript is biased to a certain point of view and the design of the questionnaire also reflects this bias. Hence, I find the current analysis to be inadequate for the task. For now, I recommend a major revision, in hope that the authors will be able to fix the issues listed below.
Major issues.
- From the presented text, it appears that authors are proponents of the widespread adoption of EMR. Throughout the text, the authors push the idea that it is not done enough. However, the aim of this work is to evaluate how useful EMR is, which has to be investigated from a neutral position. To give a specific example, line 309 in the discussion states “the level of computerization of medical records in primary care is low in China”. However, this is immediately followed by “with 58.7% of community health centres having established EMRs”. I personally find this to be quite an impressive number. The same applies to the statement in the next paragraph in line 326.
To turn such a statement into an academic conclusion, it is necessary for the authors: 1) state the criteria what is a sufficient, or high level of EMR adoption. Maybe a comparison with other countries would help here. However, a comparison of China with New Zealand having 5-million people and the Netherlands (17-million) seems to be inadequate. 2) To judge the success of EMR adoption, it is inadequate to compare the current levels of adoption with countries, which started the process earlier. If such a comparison should be made, a historical context should be taken into account. How long did it take to introduce EMR in the UK? When did the US start and how much did they achieve by now?
Alternatively, authors can tone down their concluding statements.
- The questionnaire design is biased towards a better evaluation of EMR. Line 207 states the Likert scale: “1=strongly disagree, 2=disagree, 3=slightly agree, 4=agree, 5=strongly agree”. This leaves no option for a neutral answer. Later, at line 210 if is further amplified by the data analysis: “These scores were then recorded as 0 (strongly agree, or disagree) or 1 (slightly agree, agree, strongly agree)”. With input data biased towards the higher value of EMR, there is no trust in any conclusion supporting its worth (analysis on Tables 3 and 4). As an option to resolve this issue, I suggest to repeat their analysis with an opposite bias: to record strongly agree, disagree, and slightly agree as negative scores 0, and leave only agree and strongly agree as positive evaluation 1. These conclusions, which remain valid under such an opposite bias, will likely be supported by a hypothetical unbiased questionnaire. However, this is a mild suggestion, and if the authors will find another convincing method to correct the bias, I will consider it as well.
Minor issues:
Line 132: the formula is not visible.
Fig.1 The boxes are unconnected. No “flow” in the flowchart.
Statistical methods used to process the data must be presented (e.g. in appendix). For example, there are several techniques called “chi-square test”, what exactly has been done?
Symbolic column names in Tables, such as chi-square, p, H, B, etc must be explained.
Verbal descriptions of data such as “half” or “two thirds” should be given in numbers similar to numeric descriptions that follow. For example, line 276: “About two-thirds of the respondents voted “management of chronic conditions” as the most important feature of EMRs, followed by 14% for referral arrangements and 13% for appointment scheduling.” “Two-thirds” is verbal and inaccurate, while14% and 13% are precise.
Author Response
Dear Editor,
Thank you very much for inviting us to submit a revised version of our manuscript entitled " Perceived value of electronic medical records in primary care: a national cross-sectional survey of primary care workers in mainland China" (ijerph-910857). We are grateful to the reviewers for their insightful comments, which have helped improve the manuscript significantly.
We have modified the manuscript in line with all of the reviewers’ comments in track Changes. All authors have read and approved the submitted version. Please find below our point-by-point response.
We hope this time our manuscript has met the publishing standards of the journal. We look forward to a positive reply and again thank you for handling this manuscript.
Should you have any questions, please contact us without any hesitate.
Chaojie Liu, on behalf of all authors.
Point to Point Response
Reviewer 2:
Thank you for the opportunity to review the manuscript entitled “Perceived value of electronic medical records in community health services: a national cross-sectional survey of primary care workers in mainland China”. In this work, the authors performed an impressive survey on the usage of electronic medical records (EMR) in China. Then, they performed a statistical analysis of the obtained dataset, showing the correlation between a number of factors and availability/usage of EMR. While I acknowledge how valuable is the goal of this study, the persistent narrative of the manuscript is biased to a certain point of view and the design of the questionnaire also reflects this bias. Hence, I find the current analysis to be inadequate for the task. For now, I recommend a major revision, in hope that the authors will be able to fix the issues listed below.
Response: We have revised the narrative (including relevant data analyses) to avoid bias. Please find details below.
Major issues.
- From the presented text, it appears that authors are proponents of the widespread adoption of EMR. Throughout the text, the authors push the idea that it is not done enough. However, the aim of this work is to evaluate how useful EMR is, which has to be investigated from a neutral position. To give a specific example, line 309 in the discussion states “the level of computerization of medical records in primary care is low in China”. However, this is immediately followed by “with 58.7% of community health centres having established EMRs”. I personally find this to be quite an impressive number. The same applies to the statement in the next paragraph in line 326.
To turn such a statement into an academic conclusion, it is necessary for the authors: 1) state the criteria what is a sufficient, or high level of EMR adoption. Maybe a comparison with other countries would help here. However, a comparison of China with New Zealand having 5-million people and the Netherlands (17-million) seems to be inadequate. 2) To judge the success of EMR adoption, it is inadequate to compare the current levels of adoption with countries, which started the process earlier. If such a comparison should be made, a historical context should be taken into account. How long did it take to introduce EMR in the UK? When did the US start and how much did they achieve by now?
Alternatively, authors can tone down their concluding statements.
Response: Thanks for the constructive advice. We agree and have toned down relevant statements accordingly. We have also added details in the arguments against the governmental goal in China and the contextual development in other countries.
Substantial governmental investments are often required to achieve universal adoption of EMRs as is evidenced in the developed countries [7]. This usually includes governmental certification and incentive programs to push for meaningful use of EMRs [8]. From 2001 to 2015, for example, the Canadian government spent $2 billion in EMR development, which resulted in a rapid increase in the percentage of Canadian physicians using EMRs: rising from 25% in 2007 to 75% in 2014 [9,10]. Similarly, the percentage of federally qualified health centres with an EMR system in the US increased from 40% in 2009 to 93% in 2013 [11]. (Line 317-322 on page 12 in clean version)
Several studies in Canada show that Canadian doctors used 65% of all EMR functions on average [12], fewer than 80% exclusively used EMRs to record patient care data [9], and 44% reported poor compatibility with other electronic systems [13]. (Line 367-372 on page 13 in clean version)
- The questionnaire design is biased towards a better evaluation of EMR. Line 207 states the Likert scale: “1=strongly disagree, 2=disagree, 3=slightly agree, 4=agree, 5=strongly agree”. This leaves no option for a neutral answer. Later, at line 210 if is further amplified by the data analysis: “These scores were then recorded as 0 (strongly agree, or disagree) or 1 (slightly agree, agree, strongly agree)”. With input data biased towards the higher value of EMR, there is no trust in any conclusion supporting its worth (analysis on Tables 3 and 4). As an option to resolve this issue, I suggest to repeat their analysis with an opposite bias: to record strongly agree, disagree, and slightly agree as negative scores 0, and leave only agree and strongly agree as positive evaluation 1. These conclusions, which remain valid under such an opposite bias, will likely be supported by a hypothetical unbiased questionnaire. However, this is a mild suggestion, and if the authors will find another convincing method to correct the bias, I will consider it as well.
Response: Thanks for pointing out the important issue. We agree that the merge of alternative answer categories is arbitrary and problematic. In the revised manuscript, we have maintained the original categories and performed ordinal regression analyses to correct the problem. The results remain largely the same.
Minor issues:
Line 132: the formula is not visible.
Response: corrected
Fig.1 The boxes are unconnected. No “flow” in the flowchart.
Response: corrected
Statistical methods used to process the data must be presented (e.g. in appendix). For example, there are several techniques called “chi-square test”, what exactly has been done?
Response: details added
Symbolic column names in Tables, such as chi-square, p, H, B, etc must be explained.
Response: details added
Verbal descriptions of data such as “half” or “two thirds” should be given in numbers similar to numeric descriptions that follow. For example, line 276: “About two-thirds of the respondents voted “management of chronic conditions” as the most important feature of EMRs, followed by 14% for referral arrangements and 13% for appointment scheduling.” “Two-thirds” is verbal and inaccurate, while14% and 13% are precise.
Response: corrected
Reference
- Yuefeng Li; Jianping Hu; Qun Meng. SWOT analysis on population health information development in China Chinese Journal of Health Information Management 2016, 13, 45-50 DOI: 10.3969/j.issn.1672-5166.2016.3901.3909.
- Li, D.; Wei, X.; Wu, H.; Liu, X.; Ge, C.; Gao, W. Effect of an intelligent family physician-optimised coordination model program on hypertension management in a Beijing community. Aust J Prim Health 2020, 26, 402-409, doi:10.1071/py19218.
- Xue-Juan, W.; Hao, W.; Cai-Ying, G.; Xin-Ying, L.; Hong-Yan, J.; Li, W.; Xiao-Ling, G.; Wan-Ying, L.; Wen-Juan, G.; Wan-Nian, L. Impact of an intelligent chronic disease management system on patients with type 2 diabetes mellitus in a Beijing community. BMC Health Serv Res 2018, 18, 821, doi:10.1186/s12913-018-3610-z.
- Lei, J.; Meng, Q.; Li, Y.; Liang, M.; Zheng, K. The evolution of medical informatics in China: A retrospective study and lessons learned. International Journal of Medical Informatics 2016, 92, 8-14, doi:https://doi.org/10.1016/j.ijmedinf.2016.04.011.
- Xue, Y.; Liang, H.; Wu, X.; Gong, H.; Li, B.; Zhang, Y. Effects of electronic medical record in a Chinese hospital: a time series study. Int J Med Inform 2012, 81, 683-689, doi:10.1016/j.ijmedinf.2012.05.017.
- Liu, C.; Legge, D. Challenges in China's health system reform: lessons from other countries. Aust J Prim Health 2017, 23, i-ii, doi:10.1071/PYv23n4_ED.
- O'Donnell, A.; Kaner, E.; Shaw, C.; Haighton, C. Primary care physicians' attitudes to the adoption of electronic medical records: a systematic review and evidence synthesis using the clinical adoption framework. BMC Med Inform Decis Mak 2018, 18, 101, doi:10.1186/s12911-018-0703-x.
- Raymond, L.; Pare, G.; Marchand, M. Extended use of electronic health records by primary care physicians: does the electronic health record artefact matter? Health Informatics J 2017, 25, 71-82, doi:10.1177/1460458217704244.
- Persaud, N. A national electronic health record for primary care. CMAJ 2019, 191, E28-E29, doi:10.1503/cmaj.181647.
- Canadian Institute for Health Information. How Canada Compares: Results From the Commonwealth Fund’s 2019 International Health Policy Survey of Primary Care Physicians; Ottawa, ON: CIHI: 2020.
- Ryan, J.; Doty, M.M.; Abrams, M.K.; Riley, P. The Adoption and Use of Health Information Technology by Community Health Centers, 2009-2013. Issue Brief (Commonwealth Fund) 2014, 1-8.
- Pare, G.; Raymond, L.; Guinea, A.O.; Poba-Nzaou, P.; Trudel, M.C.; Marsan, J.; Micheneau, T. Electronic health record usage behaviors in primary care medical practices: A survey of family physicians in Canada. Int J Med Inform 2015, 84, 857-867, doi:10.1016/j.ijmedinf.2015.07.005.
- Collier, R. National Physician Survey: EMR use at 75%. CMAJ : Canadian Medical Association journal = journal de l'Association medicale canadienne 2015, 187, E17-E18, doi:10.1503/cmaj.109-4957.
Round 2
Reviewer 2 Report
The updated manuscript resolves my concerns. Now, I recommend this work for publication.